# Exceptionally Uniform Bat Assemblages across Different Forest Habitats Are Dominated by Single Hyperabundant Generalist Species

**Mateusz Ciechanowski** [1,2,*], **Zuzanna Wikar** [1], **Katarzyna Borzym** [2,3], **Emilia Janikowska** [3], **Julia Brachman** [3], **Martyna Jankowska-Jarek** [1,2] **and Konrad Bidziński** [1,2]

1    Department of Vertebrate Ecology and Zoology, Faculty of Biology, University of Gdańsk, 80-308 Gdańsk, Poland; zuzanna.wikar@ug.edu.pl (Z.W.); martyna.jankowskajarek@gmail.com (M.J.-J.); konradbidzinski@gmail.com (K.B.)

2    Academic Bat Group, Polish Society for Nature Protection "Salamandra", 80-308 Gdańsk, Poland; tora.rouma@gmail.com

3    Student Bat Research Group, Faculty of Biology, University of Gdańsk, 80-308 Gdańsk, Poland; e.janikowska.519@studms.ug.edu.pl (E.J.); jubrachman@gmail.com (J.B.)

\*    Correspondence: mateusz.ciechanowski@ug.edu.pl; Tel.: +48-606-221-993

**Abstract:** Woodland bat assemblages are usually structured in a space according to the distance from the ground, water, and obstacles, features that often define chiropteran hunting tactics. Consequently, the bat species composition differs strongly among various habitats, even within the same forest patch. However, when conducting local bat surveys in Wolin National Park (WPN), we revealed an unexpected uniformity in the qualitative and quantitative structure of bat assemblages, based on mist netting and ultrasound recording. In total, 10 vespertilionid species were detected. Across all methods and sampled habitats, a single species, *Pipistrellus pygmaeus*, predominated, while no *Barbastella barbastellus*, an old forest specialist, were detected, despite the abundance of their preferred daily roosts. We also reviewed the literature for mist-netted bat samples in four different habitats in lowland Polish forests. The samples usually clustered based on habitats, and the same habitat classes often clustered very closely despite representing geographically distant forests. The exception was WPN, where all four habitat classes formed a tightly packed cluster. We hypothesize that *P. pygmaeus* might act as a hyperabundant native species, a successful generalist that reduces the contribution of more specialized taxa in the assemblage. It probably benefits from both forest renaturation and anthropogenic cross-boundary subsidy.

**Keywords:** Chiroptera; Vespertilionidae; woodlands; diversity; Poland; Wolin National Park

## 1. Introduction

Forests constitute one of the most important habitats for insectivorous bats worldwide, providing opportunities for both foraging and roosting [1]. Bats themselves might perform a significant function in the suppression of herbivorous insects and, as a consequence, drive a top-down trophic cascade in forest ecosystems [2,3]. Thus, insectivorous bats act as agents of biological pest control in silviculture [4,5]. Due to significant interspecific variation in diet among sympatric bat species [6,7], the significance of that function might be affected by the taxonomic composition of particular bat assemblages, which, in turn, is affected by a number of factors, acting on various spatial scales.

Chiropteran assemblages are structured mainly according to responses to resources and evolutionary and geological history, as well as geographic factors, while the interspecific interactions, like competition, seem to be of lesser importance [8], though they are not negligible. On the local scale, however, the proportion of particular taxa among foraging or commuting bats in a woodland landscape of the temperate zone is shaped primarily by

the spatial structure of the forest habitat [9,10], as bat species partition their niches based on the distance from the ground, water, and obstacles [11]. These species are adapted to these distances through different wing shapes, flight speed, and maneuverability [11], but also echolocation call designs [12]. The strong variation in the structure of forest bat assemblages is also caused by the tree stand age [13,14], level of natural and human-made disturbance [15], proportion of particular tree taxa [16,17], volume of deadwood [18], elevation [15], size of forest patch [19], location within the patch [20], and land cover adjacent to its borders [14]. As the bat faunal size correlates with the availability and diversity of roosts [21], while even closely related and morphologically similar species differ in roost selection [22,23], the structure of bat assemblage foraging in woodlands might be expected to result from the presence or absence of particular roost types, either within the forest or in its close neighborhood. The majority of the methods used to sample bat assemblages (mist netting and ultrasound recording) do not allow us to differentiate between the various functions of a habitat during night-time bat activity (as a foraging site, drinking site, or commuting route [24–26]), while different functions might also be expected to explain the taxonomic structures of bat samples. The most notable of such differences, hardly linked to the spatial structure of a forest habitat alone, are those between water bodies and forest roads [20] or between forest rivers and small ponds [27,28]. Irrespectively of the casual mechanisms behind the observed variation, strong differences in bat species composition are expected among different habitats, even on a relatively small geographic scale, e.g., within the same forest patch, mesoregion, or protected area.

The aim of this paper is to present the bat assemblages in a national park in Central Europe that do not meet the aforementioned expectations; i.e., they reveal an unusual uniformity in their qualitative and quantitative structures across different forest habitats and methods of sampling.

## 2. Materials and Methods

### 2.1. Study Area

Wolin National Park (WNP) is located in northwestern Poland (Figure 1), on the Baltic Sea Coast, covering a significant portion of the largest Polish island, Wolin, separated from the mainland by the Dziwna channel, which is 90 m wide at the narrowest point and ~4.5 km at the widest. The Park was created in 1960. It covers 10,937 ha, including 4648 ha of forest ecosystems (42.5%) [29]. Most of its woodlands form a compact block, covering a range of moraine hills with their culmination at 115 m a.s.l. From the north, they are bordered by active cliffs up to 93 m high, made of clay, sand, and gravel, falling directly to the Baltic Sea. Lower cliffs, also partially active, fall into the Szczecin Lagoon, bordering the Park from the south, while, in the north-east, sand dunes can be found. Four postglacial lakes are located within the park forests or adjacent to them; they are accompanied by three artificial lakes, formed in abandoned chalk pits. The total surface area of all the lakes comprises 163 ha. Except these lakes, there are only few small water bodies, all of which are shallow pools on impermeable clays, used by ungulates as drinking holes and wallows. Running waters are extremely scarce, include only four short streams, discharging the lakes, and a few even shorter creeks seeping from the southern cliff and entering the Szczecin Lagoon.

The climate, according to the Köppen classification, is a humid continental one, with a warm summer subtype (*Dbf*). The mean annual temperature is 9.1 °C, with a mean temperature in January of 0.8 °C and a mean temperature in July of 18.2 °C. The mean annual precipitation is 585 mm, with 30 days of snow cover on average.

The average age of the tree stands in WPN is 112 years. The predominant tree species are Scotch pine *Pinus sylvestris* (54.5% of the stand volume, average age 96 years), beech *Fagus sylvatica* (26.1%, 131 years), and sessile oak *Quercus petraea* (11.7%, 106 years). The predominant forest communities are fertile (*Galio odorati–Fagetum*) and acid-poor (*Luzulo pilosae–Fagetum*) beech forests, with a significant contribution of acid beech–oak *Fago–Quercetum* and suboceanic pine *Leucobryo–Pinetum* forests. However, only 26% of the

communities remain unaltered by previous human interference [30]. Intensive restoration measures have been implemented in the last few years, mostly focused on the removal of Scotch pine planted in the past in broadleaved forests. The predominantly moraine, woodland part of the Park is adjacent to the flat, alluvial part, located in the regressive delta of the Świna channel, covered predominantly by halophytic pastures and reed beds, and fed by the brackish, hypereutrophic waters of the Szczecin Lagoon.

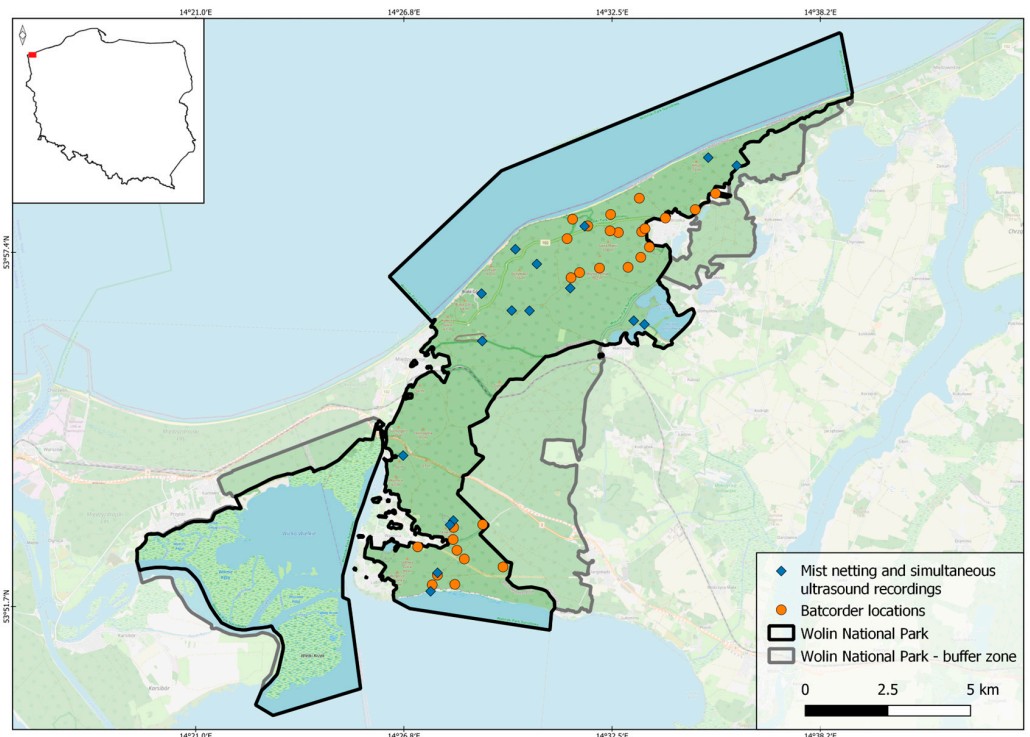

**Figure 1.** Location of the bat mist netting and ultrasound recording sites in Wolin National Park in 2022–2023.

Only a few buildings are present within the studied forest block, mostly belonging to the Park authorities (offices, living quarters, foresters' lodges, and facilities associated with the European bison breeding program). However, numerous settlements, including several villages and one city (Międzyzdroje), are located at the Park borders. The number of new buildings, ranging from hotels to small cabins, is growing fast around the Park, due to the ongoing tourist boom.

*2.2. Data Collection*

A bat survey in Wolin National Park was conducted in 2022–2023; it was originally designed to establish a basic inventory of the bat fauna, including the species composition, distribution, and location of sites of particular importance. The summer part of this survey was based on three methods.

On 10–20 July 2022–2023, bats were captured in mist nets set on forest roads, within forest stands free of understory, along forested shores of lakes and lagoons, over small brooks and streams (either free-flowing or dammed), and over small, natural forest pools (always without outflow, used as water holes by ungulates). In total, 17 sites were surveyed (18 nights, in total) (Figure 1). Every night, 3–6 monofilament mist nets (6–12 m each) were set and checked every 15 min from dusk to dawn. Bats were identified based on external morphological features, both metric and non-metric [31] and, after checking their sex, age, and reproductive status, released on the spot. All bat species known to occur in Central Europe are possible to identify unambiguously based on external morphology, sometimes combined with externally visible dental features [31], so there was neither the need to collect voucher specimens nor to resort to cranial analysis. All the nights of mist netting

were rainless, with wind speeds below 6 m/s. Bats were captured under a license from the Ministry of Climate and Environment DOP-WPN.61.122.2022.MŚP.

During the mist netting, echolocation calls were recorded automatically, using broadband, full-spectrum ultrasound detectors: Pettersson D-1000X (Pettersson Elektronik AB, Uppsala, Sweden), Song Meter Mini Bat, and Echo Meter Touch with iPad Air tablet (Wildlife Acoustics, Maynard, MA, USA) at 16 sites. Detectors were located 100–250 m from the nearest mist net, usually in the canopy gap or at the lake/lagoon shore, at the ground level, with the microphone directed to the sky at angle ~45°. Recordings in WAV format were downloaded to a personal computer and identified automatically using Wildlife Acoustics Kaleidoscope Pro 5 software that included automatic classifiers for all European bat species, based on a maximum-likelihood estimator [32]. We chose the Balanced (0) level of sensitivity for the Auto ID function. About 17% of all recordings were identified manually using the BatSound 3.31 program, based on the shape of the sonograms and oscillograms, frequency of maximum amplitude, call duration, interval length, and repetition rate, compared to the values for these parameters available in the literature [31,33].

From April to October 2022, Batcorder (ecoObs GmbH, Nürnberg, Germany) ultrasound recorders were set for 1–5 nights (median 2) by members of WPN staff at 31 sites (88 nights in total) (Figure 1). All recording points were located inside the tree stands. Recorders were hung on trees at height ~1.5–2.0 m each night (from dusk to dawn) and removed during the day. Recorded calls in RAW format were identified automatically using batIdent 1.5 software [34]. If automatic ID software recognized some files as containing call sequences of the western barbastelle *Barbastella barbastellus*, then we checked all files recorded in a particular site manually, using bcAdmin 4.0 software, to search for recordings of that particular species. We applied two different sets of bioacoustic programs, as various brands of ultrasound detectors recorded calls in one of the two aforementioned file formats, compatible only with dedicated software (WAV—BatSound 3.31 and Kaleidoscope Pro 5, RAW—batIdent 1.5 and bcAdmin 4.0).

Due to the low probability of correct identification [35], we left calls of representatives of the *Myotis* genus unidentified at the species level, with exception of *M. myotis* and *M. nattereri* [31]. If automatic classifiers identified some call sequences as having been produced by species that had never been recorded in northwestern Poland based on hand-examined specimens, then we reclassified them to the species with the closest call characteristics (*Miniopterus schreibersii* to *Pipistrellus pygmaeus*, *P. kuhlii* and *Hypsugo savii* to *P. nathusii*, *Nyctalus lasiopterus* to *N. noctula*). The genera *Nyctalus*, *Eptesicus*, and *Vespertilio* are also known for a strong overlap in call parameters, with the pair *N. leisleri* and *V. murinus* being especially difficult to distinguish [33]. Thus, we left only automatic identifications belonging to the species from that group, whose presence was confirmed via captures in mist nets, while the remaining ones were reclassified into the broader category of *Nyctalus/Eptesicus/Vespertilio* (NEV) (Table S1).

The location of every mist net or ultrasound recorder was recorded via Locus Map application on an Android mobile phone. The classification of each location into the general habitat categories (water bodies: 1—forest-crossing stream, 2—lake or lagoon shore, lined by forest, 3—small forest pool; land: 4—canopy gap or clearing, 5—glade, 6—forest road) was completed during field work. Categories 1 and 2 represent mostly closed-canopy sites; 2, 4, and 5 represent open canopy or no canopy at all, with only the undergrowth or forest providing some clutter; while category 3 represents both open—(1) and closed-canopy (2) sites. The sites with Batcorders represented mostly closed, cluttered habitats.

### 2.3. Statistical Analysis

Particular bat species differ strongly in sonar range and detectability using ultrasonic microphones, which results in the underrepresentation of species producing low-intensity calls [33]. Therefore, we provided two measures of their relative abundance based on data obtained with ultrasound detectors. One is the raw number and percentage of echolocation call sequences classified into particular taxa. The second comprises the adjusted numbers

and the resulting percentage obtained by multiplying raw data by detectability coefficients, provided in [33], for an open-to-semi-open environment, as no recording site was located in dense forest understory (Table 1). We decided, however, to also provide the raw, unadjusted data, as most studies have not adjusted their recording numbers for detectability, while we aimed not only to compare our datasets across habitats and methods, but also to compare the structure of forest bat assemblage in WPN with those revealed in previous papers.

**Table 1.** Coefficients used to adjust the numbers of bat passes (call sequences) belonging to particular species in open-to-semi-open environments [33].

| Taxon | Detectability Coefficient |
|---|---|
| *Myotis myotis* | 1.25 |
| Small *Myotis* [1] | 1.67 |
| *Eptesicus serotinus* | 0.63 |
| *Pipistrellus* spp. | 1.00 |
| *Nyctalus noctula* | 0.25 |
| *Nyctalus leisleri* | 0.31 |
| *Plecotus auritus* | 1.25 |

[1] *M. nattereri* or any representative of *M. daubentonii/mystacinus/brandtii*, classified as *Myotis* sp.

We used capture and acoustic data separately to compute the expected species richness, based on the individual abundance rarefaction (species accumulation curve), using PAST ver. 4.07b [36]. This method allowed the in-detail comparison of bat species diversity among habitats. We also compared the bat assemblages of WPN mist-netted on forest roads, over streams and small bodies of water, and at lakes and lagoons with those netted in other forests in the lowland part of Poland in 1990–2023, classified into similar habitats (1—rivers and streams, 2—lakes, lagoons and large fish ponds, 3—small ponds, 4—forest roads). To perform such a comparison, we reviewed all the already-published data on bat surveys where bats were mist-netted in summer (Table S2). As the morphological features to distinguish *P. pygmaeus* from *P. pipistrellus* only became known in the early 2000s [37,38], individuals classified as either species were placed in the same category (Ppip/Ppyg), similarly to *Myotis mystacinus* and *Myotis alcathoe* (Mmys/Malc). We rejected studies where the attribution of sites to habitat classes remained ambiguous. To compare the species composition of samples among forest complexes and habitats, we performed cluster analysis in ClustVis, an open-access web tool for visualizing the clustering of multivariate data [39], provided by the University of Tartu [40]. Samples characterized by relative abundances of 17 bat species were clustered with the complete linkage method, using the two farthest objects from the two clusters to be merged, with the Euclidean distance as a measure of similarity and 'tightest cluster first' as the method of tree ordering.

## 3. Results

### 3.1. General Composition of Bat Fauna

In total, we captured 455 bats in mist nets. They represented 10 species: the greater mouse-eared bat *Myotis myotis*, Natterer's bat *Myotis nattereri*, Daubenton's bat *Myotis daubentonii*, the serotine *Eptesicus serotinus*, the common pipistrelle *Pipistrellus pipistrellus*, the soprano pipistrelle *Pipistrellus pygmaeus*, Nathusius' pipistrelle *Pipistrellus nathusii*, the noctule *Nyctalus noctula*, the lesser noctule *Nyctalus leisleri*, and the brown long-eared bat *Plecotus auritus*. We recorded all the same species with ultrasound detectors, although *M. daubentonii* could not be reliably distinguished from similar taxa (and is thus classified to *Myotis* sp. only). Although the batIdent software recognized 12 call sequences of *Barbastella barbastellus* recorded using Batcorder detectors, we did not confirm this identification via subsequent manual analysis. Pettersson D-1000X, Mini Bat (MB), and Echo Meter Touch (EMT) detectors recorded 6828 echolocation call sequences, while Batcorders recorded 14,543 such sequences.

### 3.2. Species Composition Revealed Using Different Methods

Both in the netted sample and the two datasets obtained via ultrasound recording, *P. pygmaeus* predominated most strongly among the netted individuals (60%). Specimens were captured at 14 of 17 sites, meaning that it was the most frequent species, and it was the most abundant netted bat at 13 sites. Its calls were also recorded at the highest number of sites, and at 14 sites, they appeared to be the most numerous. The distinguishing features of the netted sample were the relatively high percentage of *N. noctula* and *M. daubentonii* and the frequency of occurrence of *M. nattereri* being almost as high (13 sites) as that of *P. pygmaeus*, although *M. nattereri* was only an accessory species in the assemblage (5.1%). Batcorders distinguished from the remaining ultrasound detectors by recording unusually low number of *N. noctula* calls (3.93% of all sequences). On the contrary, among the recordings obtained with D-1000X, MB, and EMT detectors, *N. noctula* appeared to be a subdominant species (29.3%, compared to the 36.9% of *P. pygmaeus*); however, when we adjusted the results for detectability, its share in the assemblage dropped to just 8.6%. In the adjusted results, *Myotis* sp. replaced *N. noctula* in the position of the second-most-abundant taxon (27.5%) (Table 2).

**Table 2.** Abundance (n), dominance (%), and the number of localities (NL) of particular bat species in three datasets used to characterize the summer bat assemblage in Wolin National Park. NEV—*Nyctalus / Eptesicus / Vespertilio*, indet.—unidentified. MB—Mini Bat, EMT—Echo Meter Touch.

| Species | Mist Netting | | | D-1000X, MB, and EMT Detectors | | | | | Batcorder Detectors | | | | |
| | | | | Raw | | Adjusted | | | Raw | | Adjusted | | |
| | *n* | % | NL | *n* | % | *n* | % | NL | *n* | % | *n* | % | NL |
|---|---|---|---|---|---|---|---|---|---|---|---|---|---|
| *M. myotis* | 1 | 0.2 | 1 | 4 | 0.06 | 5 | 0.09 | 3 | 3 | 0.02 | 4 | 0.03 | 2 |
| *M. nattereri* | 23 | 5.1 | 13 | 3 | 0.04 | 5 | 0.09 | 2 | 20 | 0.14 | 33 | 0.23 | 6 |
| *M. daubentonii* | 51 | 11.2 | 8 | - | - | - | - | - | - | - | - | - | - |
| *Myotis* sp. | - | - | - | 952 | 14.04 | 1590 | 27.46 | 9 | 869 | 5.98 | 1451 | 10.03 | 22 |
| *E. serotinus* | 5 | 1.1 | 4 | 156 | 2.30 | 98 | 1.70 | 10 | 81 | 0.56 | 51 | 0.35 | 12 |
| *P. pipistrellus* | 5 | 1.1 | 3 | 554 | 8.17 | 554 | 9.57 | 13 | 840 | 5.78 | 840 | 5.81 | 23 |
| *P. pygmaeus* | 273 | 60.0 | 14 | 2501 | 36.89 | 2501 | 43.19 | 16 | 7724 | 53.11 | 7724 | 53.40 | 28 |
| *P. nathusii* | 12 | 2.6 | 9 | 460 | 6.78 | 460 | 7.94 | 15 | 1075 | 7.39 | 1075 | 7.43 | 26 |
| *Pipistrellus* sp. | 2 | 0.4 | 1 | 38 | 0.56 | 38 | 0.66 | 6 | 3059 | 21.03 | 3059 | 21.15 | 30 |
| *N. noctula* | 59 | 13.0 | 5 | 1988 | 29.32 | 497 | 8.58 | 12 | 853 | 5.87 | 213 | 1.47 | 22 |
| *N. leisleri* | 6 | 1.3 | 1 | 120 | 1.77 | 37 | 0.64 | 8 | 12 | 0.08 | 4 | 0.03 | 5 |
| *P. auritus* | 18 | 4.0 | 4 | 4 | 0.06 | 5 | 0.09 | 3 | 7 | 0.05 | 9 | 0.06 | 5 |
| Total | 455 | 100.0 | 17 | 6780 | 100.00 | 5790 | 100.00 | 16 | 14,543 | 100.00 | 14,463 | 100.00 | 30 |
| NEV | - | - | - | 18 | - | - | - | - | 646 | - | - | - | - |
| Indet. | - | - | - | 30 | - | - | - | - | 1619 | - | - | - | - |

### 3.3. Structures of Bat Assemblages in Different Habitats

*P. pygmaeus* was a dominant species in all the habitats sampled with mist netting, comprising almost the same percentage of individuals on forest roads and over small standing water bodies and streams (62–65%), while amounting to 39% of netted bats at lake and lagoon shores. Habitats differed in the quantitative structures of netted bat samples mostly in the abundance of subdominant and accessory species, most notably *N. noctula* in forests and at lake shores, *M. daubentonii* over streams and at lake shores, and *P. auritus* over small forest pools. The only *N. leisleri* individuals were netted over one, small forest pool (Figure 2). The species accumulation curves revealed no differences in diversity among habitats where netting was conducted, with nearly identical trends and almost-completely overlapping confidence intervals (Figure 3).

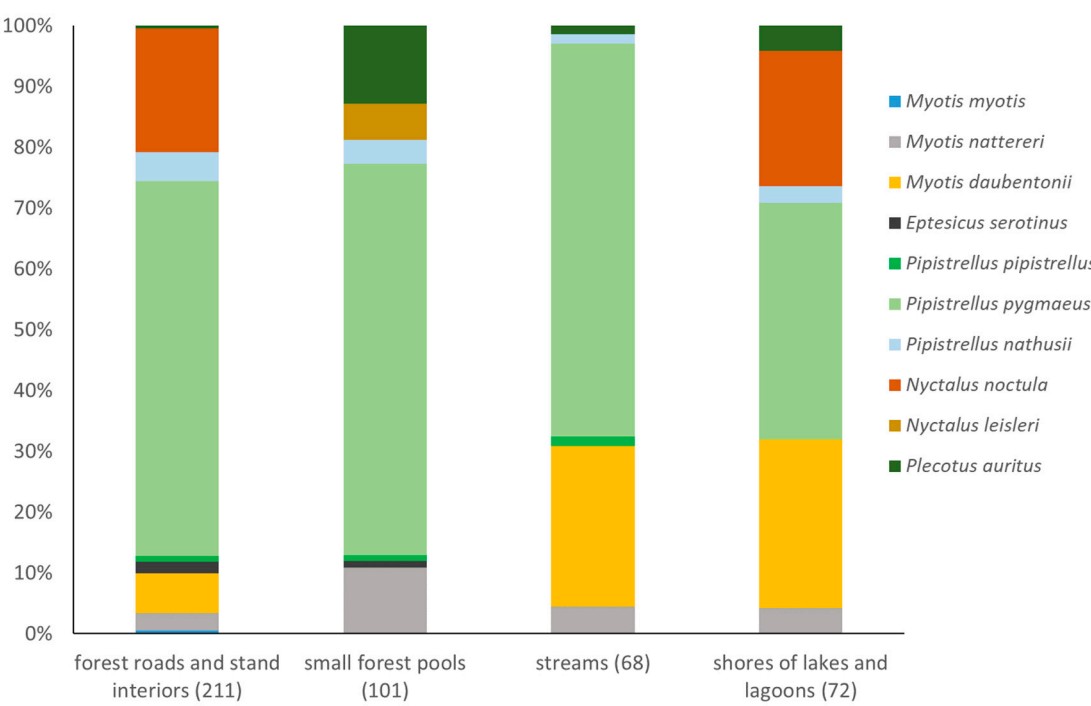

**Figure 2.** Structures of bat assemblages in four habitats within the forests of Wolin National Park, as revealed via mist netting. The sample size (the number of captures) is in brackets.

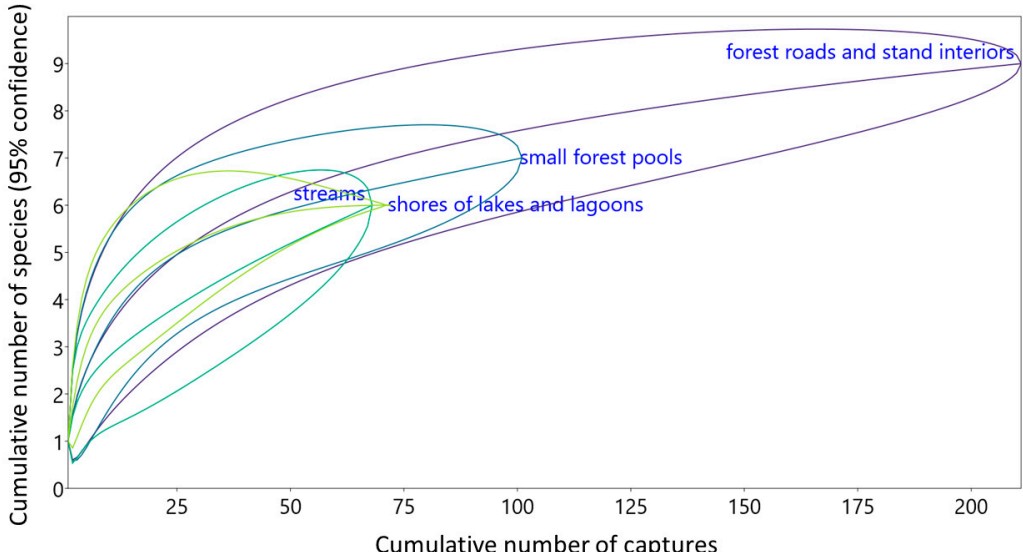

**Figure 3.** Rarefaction curves for the cumulative numbers of species plotted against the numbers of bats mist-netted in four habitats within the forests of Wolin National Park.

Ultrasound call recordings, conducted during mist netting, revealed a similar pattern. *P. pygmaeus* was the most-abundant species over canopy gaps (60%) and glades (49%), but it was the second-most-abundant species at lake shores (31%), surpassed by *N. noctula* (36%). Notable subdominant taxa were *Myotis* sp. at lake shores and *N. leisleri* over glades (Figure 4). The difference between habitats diminished even further when the data were adjusted for detectability. *P. pygmaeus* became the most-abundant species in all three habitat classes (37–69%), while *Myotis* sp. became the second-most-abundant taxon at lake shores (34%) (Figure 5). Among the recorded representatives of the genus *Pipistrellus*, a relatively high abundance of *P. nathusii* over canopy gaps and clearings is worth noting, based on the raw (16%, Figure 4) and adjusted data (17%, Figure 5). The species accumulation curves

reveal a negligibly higher diversity of bats flying in canopy gaps/clearings and virtually no differences between glades and lake shores, but the confidence intervals overlap strongly in the cases of all three habitat classes (Figure 6).

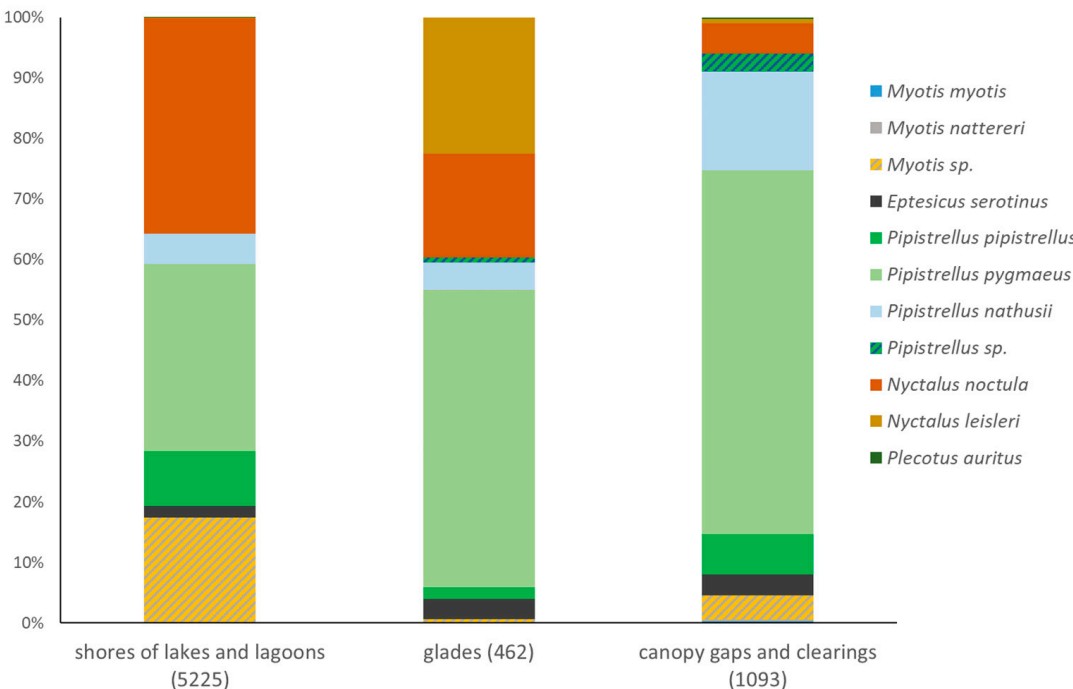

**Figure 4.** Structures of bat assemblages in three habitats within the forests of Wolin National Park, as revealed via ultrasound recordings (D-1000X, Mini Bat, and Echo Meter Touch devices, exclusively). The sample size (the raw number of echolocation call sequences) is in brackets.

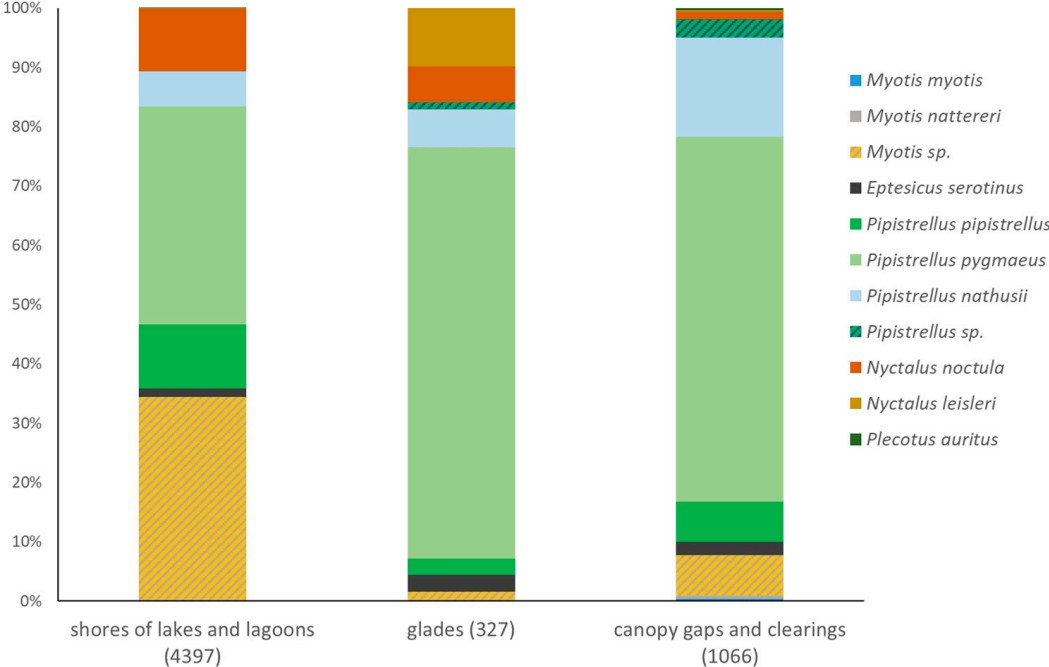

**Figure 5.** Structures of bat assemblages in three habitats within the forests of Wolin National Park, as revealed via ultrasound recordings (D-1000X, Mini Bat, and Echo Meter Touch devices, exclusively), with the proportion of each species adjusted for detectability by multiplying the numbers of call sequences by coefficients from Table 1.

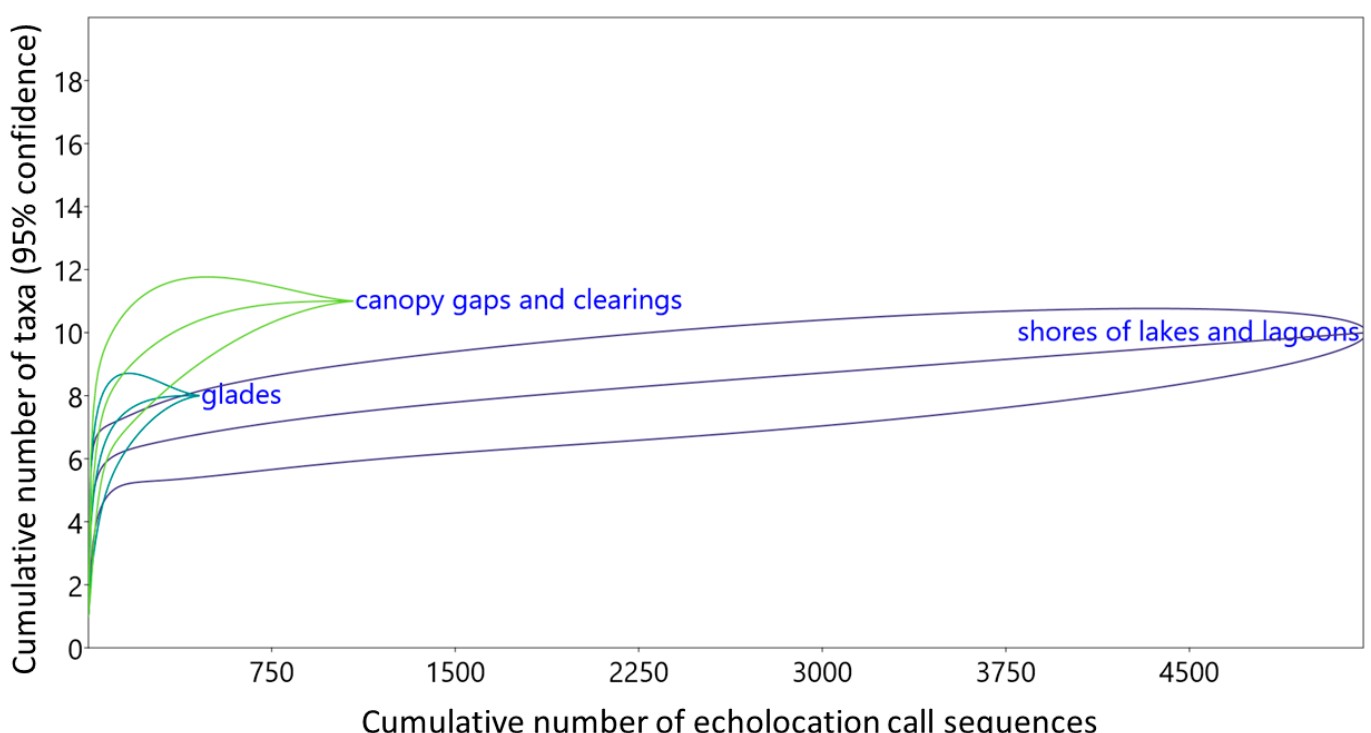

**Figure 6.** Rarefaction curves for cumulative numbers of species plotted against the raw numbers of bat call sequences recorded in three habitats within the forests of Wolin National Park (using D-1000X, Mini Bat, and Echo Meter Touch ultrasound recorders, exclusively).

*3.4. Comparison with the Other Polish Lowland Forests*

A review of the Polish literature on the fauna provided 33 papers, allowing us to retrieve data regarding bat species composition and habitats from 29 lowland forests. Combined with our data from WPN, these papers provided 55 samples, containing 5866 individual captures in total. The number of captures varied strongly, from 7 to 552 (median 70). More than one habitat class was sampled in 17 forests (Table S2). Samples were usually, although not exclusively, clustered based on habitats, with a notably high proportion of *M. daubentonii* and (regionally) *N. noctula* netted over rivers but a high proportion of *P. auritus*, *E. serotinus* and (locally) *B. barbastellus* on forest roads. Lakes and other large water bodies were nested among rivers, revealing very similar structures of bat assemblages, while small ponds represented the most diverse set of data, usually clustering closer to roads than other water bodies. The same habitat classes often clustered very closely, despite representing geographically distant forests (e.g., rivers in the Białowieża Primeval and Kozienicka Forests or roads in Strzeleckie Forests and Meteoryt Morasko Nature Reserve, located 250 km and 500 km apart, respectively). The exception was WPN, where all four habitat classes formed a tightly packed cluster, distinguished by an unusually high proportion of the *P. pipistrellus/P. pygmaeus* species group, revealed in no other location. In no other forest were all the sampled habitats clustered together, while the two habitat classes from the same forest clustered together only in three cases (Figure 7).

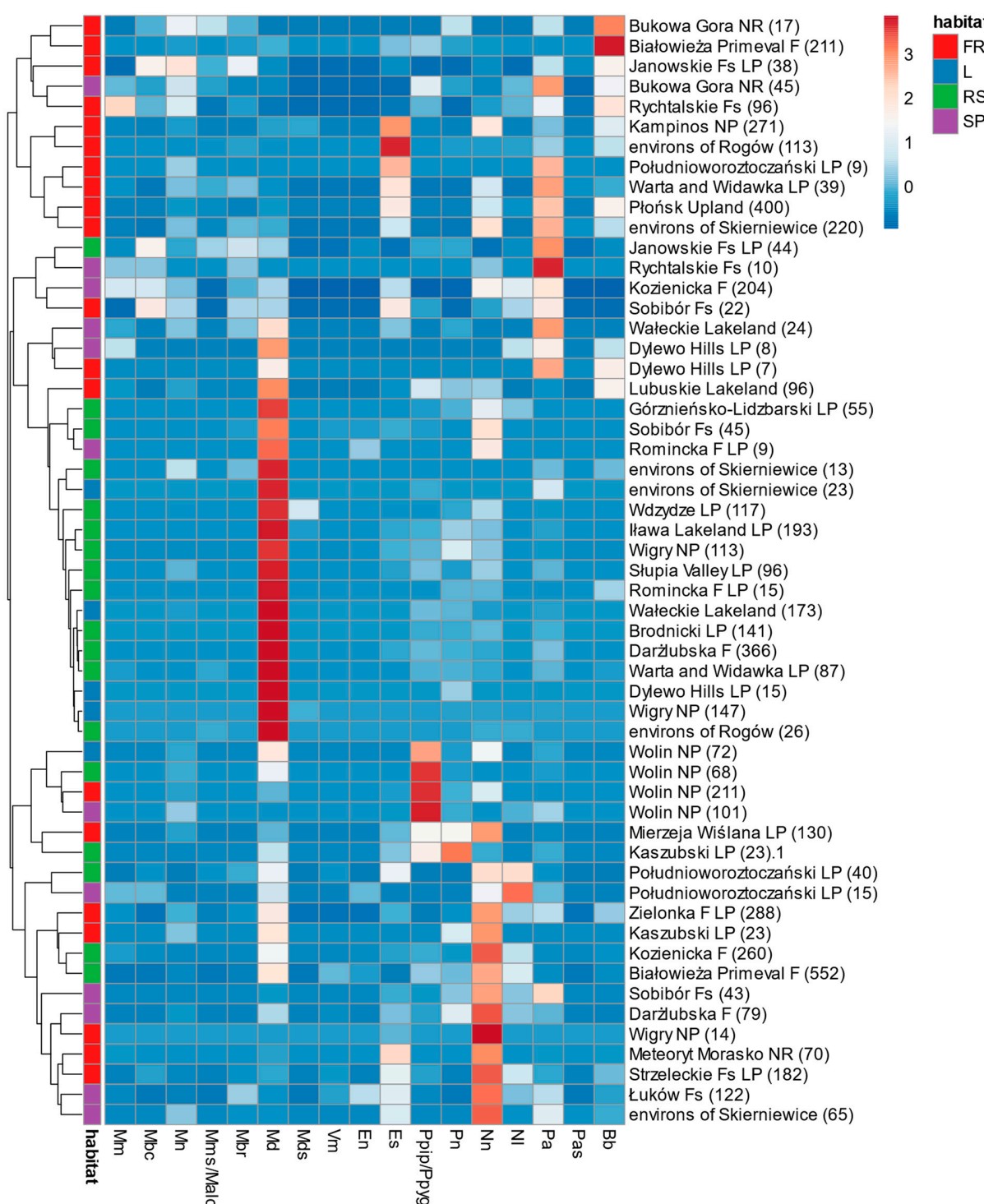

**Figure 7.** Comparison of species composition of bats mist-netted in lowland Polish forests, based on the literature analysis (Table S2), subjected to cluster analysis (clustering method—complete linkage, Euclidean distance, tightest cluster first). Habitats: RS—rivers and streams, L—lakes, large fishponds, and lagoons, SP—small ponds, FR—forest roads and tree stand interiors. Species: Mm—*Myotis myotis*, Mbc—*M. bechsteinii*, Mn—*M. nattereri*, Mms/Malc—*M. mystacinus/alcathoe*, Mbr—*M. brandtii*, Md—*M. daubentonii*, Mds—*M. dasycneme*, Vm—*Vespertilio murinus*, En—*Eptesicus nilssonii*, Es—*E. serotinus*, Ppip/Ppyg—*Pipistrellus pipistrellus/pygmaeus*, Pn—*P. nathusii*, Nn—*Nyctalus noctula*, Nl—*N. leisleri*, Pa—*Plecotus auritus*, Pas—*P. austriacus*, Bb—*Barbastella barbastellus*. Site names: F—forest, Fs—forests, NP—national park, LP—landscape park, NR—nature reserve. The sample sizes (numbers of captures) are in brackets.

## 4. Discussion

### 4.1. General Composition of Bat Fauna

The summer assemblage of bats inhabiting the forests of WPN appeared to consist of 10 species, belonging to 5 genera and a single family, Vespertilionidae. These species represent five foraging guilds [8]: open-air aerial foragers (*N. noctula*, *N. leisleri*), forest and clearing aerial foragers (*E. serotinus*, *P. pipistrellus*, *P. pygmaeus*, *P. nathusii*), water-surface foragers (*M. daubentonii*), foliage gleaners (*M. nattereri*, *P. auritus*), and a ground gleaner (*M. myotis*). The taxonomic and ecological composition of the assemblage is typical of lowland Central or Northern Europe, similar to that recorded in southern Sweden or Germany [8].

However, not all species whose geographical ranges cover WPN (i.e., they occur in northwestern Poland [41]) nor even all the species known to occur within the borders of the Park were recorded during our survey. Two species were netted in WPN solely during autumn, swarming at abandoned post-WW2 bunkers; these were Brandt's bats *Myotis brandtii* and western barbastelles *B. barbastellus* [42]. As both species are regional or facultative migrants, revealing moderate but regular movements between summer and winter areas [43,44], it is quite possible that they do not occur in summer on the island of Wolin at all. A previous acoustic survey in WPN, based solely on Batcorder detectors, provided some scarce recordings of *B. barbastellus*; however, the authors used exclusively automatic identification and did not verify any IDs manually [45]. By verifying recorded files manually, we provided evidence that batIdent software alone is not a reliable method for the detection of *B. barbastellus* during acoustic surveys due to widespread misidentification. The only published field test, comparing the accuracy of identification for software used in our study (batIdent and Kaleidoscope) unfortunately did not cover *B. barbastellus* [44].

The parti-colored bat *Vespertilio murinus* was recorded more than 100 years ago on the neighboring island of Uznam, about 10 km from the study area, based on collected specimens [46]. Its echolocation calls were identified using batIdent software during both our and earlier Batcorder surveys [45]; however, relying exclusively on auto ID is even less reliable than in the case of the western barbastelle. *V. murinus* is a species widely known for extreme call variation [47], revealing an almost-complete overlap of all call parameters with four other sympatric species, i.e., *N. noctula*, *N. leisleri*, *E. serotinus*, and *E. nilssonii* [48]. The performance of both batIdent and Kaleidoscope in recognition of *V. murinus* is generally low (66.7% and 16.7% of correct IDs, respectively, [44]); thus, we refrained from including this species in our database. As *V. murinus* is a long-distance migrant [44], regularly using coastal and insular areas of the southern Baltic Sea during its spring and autumn movements [49–51], it is highly probable that it visits WPN as well, but we cannot provide any reliable evidence that it contributes to the summer bat assemblage in that area. A relatively large number of capture data (the third-largest netting sample from any lowland forest in Poland—Table S2), with not even a single *V. murinus* caught, makes its occurrence in WPN during summer unlikely. Three other species, *Myotis bechsteinii*, *M. dasycneme*, and *E. nilssonii*, do also occur in northwestern Poland [41], known at sites located 50–100 km south of WPN [52–54]. Their absence within the borders of the Park is also notable.

### 4.2. Species Composition Revealed Using Different Methods

Both mist nets and ultrasound detectors can yield biased samples of bat assemblages, resulting in different proportions of detected species revealed using either method [55,56]; thus, the two methods are often considered complementary [57]. Mist nets tend to underrepresent species with extremely high flight altitude, high maneuverability, and/or high sonar resolution, while ultrasound detectors tend to underrepresent species with low-intensity sonar [56]. Some of these features may seem mutually exclusive; e.g., fast-flying species with low maneuverability, like *N. noctula*, also hunt at high altitude [11], usually above mist nets, but they can be effectively netted at drinking sites with low clutter [28,58], where they need to decrease their distance from the ground level. Among the

highly maneuverable species, water-surface foragers, like *M. daubentonii*, were considered overrepresented, probably due to their significantly lower flight altitude [59], compared to aerial-hawking pipistrelles [28]. Some of the differences revealed via mist netting at water bodies hardly apply to mist netting on commuting routes, like forest roads, however. The increased popularity of monofilament (ultra-thin) mist nets, replacing the thicker polyester or nylon nets that were predominant in earlier studies, may partially reduce bias in capture data and allow the capture of even those insectivorous bats with the highest sonar resolution [60]. In our study, the application of monofilament nets allowed us to effectively sample even *M. nattereri*, the species adapted to the smallest target range resolution among the representatives of the genus *Myotis* and, as a result, able to effectively navigate even a few centimeters from obstacles [61].

Bias in acoustic surveys is much easier to quantify, as it is linked almost exclusively to the call amplitude and resulting sonar range, at least if the species is recognizable based on recordings alone [56]. It explains the much higher proportion of *P. auritus*, emitting low-intensity calls and often relying on passive listening, in our netted sample and the much higher proportion of *N. noctula*, emitting the loudest calls among bats in WPN, in parallel-collected acoustic samples. This bias might be adjusted for detectability using species-specific coefficients [33], as we did in our study. Although the application of these coefficients reduced the bias toward the loudest species, it did not change the overall picture of the bat assemblage, with *P. pygmaeus* predominating in all three methods. No similar coefficients can be designed for the detectability of species with mist netting at present. Still, a significant portion of species cannot be recognized based on echolocation calls, due to a strong overlap in call parameters and, in some groups (e.g., the genus *Myotis*), this problem cannot be entirely resolved, even with the application of discriminant function analysis [62] or artificial neural networks [63]. It also applies to our survey, as our automatic classifiers worked based on similar principles, and we had to confirm the presence of the most numerous representative of the genus *Myotis* (*M. daubentonii*) with capture data. Similarly to how mist nets are made of mesh of different thicknesses, various models of bat detectors may also differ strongly in sensitivity, and these differences are not evenly distributed among call frequencies [64,65]. Additional bias may result from the fact that—at least at close-canopy sites—some high-flying species, like noctules, may not even represent the habitat in which their calls were recorded, as they often use the open space above the canopy but their calls easily reach the microphone located at ground level. It may lead to the further overrepresentation of some taxa that are already overrepresented due to high-intensity calls.

### 4.3. Uniformization of Bat Assemblages across Habitats—A Unique Feature of Wolin National Park?

The available acoustic data on forest bat assemblages in lowland Poland are scarce, restricted to just a few woodland areas [13,17,66–68], making broadscale comparisons, including cluster analysis, virtually impossible. Thus, we restricted our clustering test solely to capture data, as they are available in numerous locations in all geographical regions, usually collected as a basis for local bat surveys (Table S2). Usually, local bat assemblages clustered based on habitat affiliation. Both running and large stagnant waters share a predominance of *M. daubentonii*, a species that is a specialist water-surface forager [11]. Thus, only a large surface of open water can support a nursery colony in its vicinity; such a function could be played not only by a lake or a group of fishponds but also by a river, even a small one, as, along the longer section, it may provide a large area of water as well [28,69]. Locally, *N. noctula* and/or *N. leisleri* also appear as indicators of forest rivers are hardly universal features of Polish bat assemblages. River valleys, where noctules predominate, are covered mostly by open meadows [58], unlike many riverine sites where only *M. daubentonii* predominate and river channels are covered by the canopies of riparian woodlands [28]. Meanwhile, noctules—hawking prey highly above the ground, and presumably flying

down to the water only to drink—require open space, as their fast, unmaneuverable flight precludes them from flying close to obstacles, including vegetation [11].

The divergence of bat assemblages visiting small forest ponds, compared to other water bodies, is probably a result of their specific function—they are not foraging but drinking sites. This is why they attract a number of species that do not hunt over water at all, especially gleaners (*Plecotus* spp., *M. myotis, M. bechsteinii, M. nattereri* [11,31]), but also, in some regions, large aerial hawkers that may sometimes hunt over water opportunistically (genera *Nyctalus* and *Eptesicus* [67]) but primarily visit water sources to drink. As forest roads act both as commuting routes and as foraging sites [24], offering spaces with reduced clutter, they appear also to have been used by members of various foraging guilds. The latter is probably the reason why assemblages of bats using forest roads often cluster together with small forest ponds. Despite having different functions, they aggregate a set of diverse species, revealing not only diverse hunting strategies but also diverse roost preferences; the indicator species of this habitat include obligatory house dwellers (*E. serotinus*), old forest specialists (*B. barbastellus*), and generalists (*P. auritus*) [31]. However, there are some regions of Poland where *N. noctula* can also predominate in samples netted on forest roads; this might be explained by the proximity of daily roosts, as this species is also a tree dweller, selecting shelters in sites with reduced clutter [22].

Contrary to the general pattern of clustering similar habitats first, the netted samples from WPN formed a close cluster based on geographic location, a unique position among all the analyzed data from lowland Poland, resulting from the domination of a single species, *P. pygmaeus*. Notable, secondary differences between habitats were revealed, and they partially followed the same pattern, as revealed for the remaining Polish forests (a higher abundance of *P. auritus* on small pools, or *M. daubentonii* over streams and larger waters). These differences were not strong enough to overcome the effect of pipistrelles predominating in every habitat. The high abundance and frequency of occurrence of bats from the genus *Pipistrellus* is specific to northern Poland, especially the lakelands and Baltic Sea coast, compared to the southern part of the country [41]. That differences are suggested to be the result of reduced competition with the *Myotis mystacinus* species complex, which is extremely rare in the north but numerous in the south [70]. This does not explain, however, the domination of a single species in all the sampled habitats, despite the application of various methods and does not explain why, among all Central European pipistrelles, the species occupying the position of most dominant is *P. pygmaeus*.

However, it is impossible to check whether the unusual species composition of bats in WPN is a result of recent changes or whether it has been a feature of this region for a longer time. The only other systematic bat survey of the Park of which the quantitative data have ever been published was restricted to one method (an acoustic survey with Batcorders) and one class of habitats (forest roads and the interior of tree stands). Yet, it revealed the predominance of *P. pygmaeus* at all sampled sites; however, these data were collected only 4–5 years before our study [45].

### 4.4. Factors behind the Hyperabundance of Pipistrellus Pygmaeus and the Scarcity of Forest Specialists

To our knowledge, the only other forest in the Central European nemoral zone where *P. pygmaeus* has been recorded as dominating in all sampled habitats is Włoszczakowice Forest District in western Poland. *P. pygmaeus* accounted for 42–48% of echolocation call sequences there, depending on the forest type [17]. The species might, however, predominate in some extremely depauperated bat assemblages, for example, in northeast Scotland [71]. Both areas were sampled using acoustic surveys alone, and in no other region did *P. pygmaeus* predominate, either among netted bats or in all the studied habitats. In Białowieża Primeval Forest, it was the bat most frequently recorded in the strict protection zone of the national park, but it was outnumbered by *B. barbastellus* among recordings from the part of that forest that was subjected to logging and among the bats netted on forest roads [72]. In the Gdańsk Pomerania region, also located in northern Poland but further

east, *P. pygmaeus* appeared to be the rarest among three species of pipistrelles, outnumbered not only by *P. pipistrellus* and *P. nathusii* but also by *N. noctula*, *E. serotinus*, and *Myotis* sp. (69 sites, N = 4063 call sequences, [67]).

Pipistrelle bats are considered generalists, representing the *r* selection model on the mboxemphr–*K* strategy spectrum among chiropteran life histories, due to their relatively small body size, short life, and high fecundity and dispersal [73]. They can thrive in variable or unpredictable environments and outcompete more specialized species via exploitation, which has already been suggested as the mechanism explaining the decline of the threatened lesser horseshoe bat, *Rhinolophus hipposideros*, following the expansion of *P. pipistrellus* in Switzerland, due to their complete overlap in spring diet [74]. Thus, we could expect their anthropogenically driven expansion to lead to the depauperation of local bat fauna. Their ecological plasticity allows them to predominate in urban and agricultural landscapes, as revealed via acoustic surveys [67,75,76], although, as revealed in our study, they appear never to predominate among netted bats in forests (Table S2). However, within the *Pipistrellus pipistrellus* species complex, *P. pygmaeus* is always considered a relative specialist regarding its habitat and landscape affiliation. It occupies a much narrower habitat niche, requiring the presence of wetlands, a developed riparian zone, and broadleaved forests, while its sibling taxon, *P. pipistrellus*, appears to be eurytopic, widely utilizing an agricultural landscape beyond water [67,77–79].

Nonetheless, the classification of *P. pygmaeus* as 'specialist' relies solely on the habitats used as foraging sites. The remaining aspects of its biology reveal its enormous ecological plasticity and resistance to anthropopressure. The species uses an unusual variety of nursery roosts, from buildings [80] to hollow trees [81,82], spaces under exfoliating bark [82], bat boxes [81], and bridges [83], while its house-dwelling behavior reveals no selectivity regarding building age, material, or structure [84]. It also forms nursery colonies much larger than those of *P. pipistrellus* [85], furtherly increasing its competitive potential. Moreover, due to flexible foraging strategies, *P. pygmaeus* is adapted to exploit abundant but ephemeral prey, such as small midges, forming large swarms around even-aged, intensively managed plantations of non-native conifers [86]. At least when foraging in riparian and wetland habitats, *P. pygmaeus* hunts mostly small dipterans belonging to the family Chironomidae, i.e., one of the most abundant groups of aquatic insects, which often increase in density through the anthropogenic eutrophication of waters [87]. Finally, the species appears not to be affected by exclusion from roosts, easily switching to alternative daily shelters and resuming their former activities and patterns [80].

Therefore, the roosting and foraging opportunities in forests in WPN appear to form a set of conditions able to maintain an unusually high abundance of *P. pygmaeus*, to the point that it numerically predominates in all bat assemblages. Firstly, the native beech forests of the park, due to consistent strict protection, are undergoing a constant increase in stand age and an abundance of dead and dying trees, providing loose bark and crevices, usually preferred by *P. pygmaeus* [82], but also the appearance of canopy gaps. On the other hand, the stand conversion conducted in forest communities with an anthropogenically altered structure (based on the selective cutting of formerly planted Scotch pine) led to an increase in the abundance of coarse woody debris and the appearance of numerous pine snags with vertical splits and loose bark, which were left to naturally decay. It also resulted in the thinning of the whole stand and the appearance of several small clearings. Both processes—natural and anthropogenic—can be considered to comprise a broad-scale disturbance in the forest ecosystems of WPN, and such disturbances are known to increase the abundance of *P. pygmaeus*, as was exemplified by the outbreak of bark beetles *Ips typographus* in Białowieża Primeval Forest [68]. The processes that lead to the appearance of numerous snags and deadwood should, however, also favor old-forest specialists like *B. barbastellus* [88], which is also known to select roosts under loose bark [89], or *N. leisleri*, which selects rot cavities [22]. This is not the case in WPN, however, where *B. barbastellus* seems not to occur in summer at all, despite there being large areas of old beech forests, its optimal habitat [89], while *N. leisleri* seems to be a scarce and local

species. A possible explanation for such high dominance indices of *P. pygmaeus* could be cross-boundary subsidy in both surplus roosts and prey, which act in synergy with the processes affecting forests within WPN. Firstly, the chaotic urbanization of areas located at the very borders of the Park, as well as the unorganized expansion of recreational development in more rural parts of WPN's neighborhood, may provide abundant new daily shelters. As *P. pygmaeus* can easily inhabit even new buildings [84] and is able to enter even very narrow crevices, it can benefit rapidly from ongoing urbanization [90]. However, the crucial subsidy that may contribute to the hyperabundance of *P. pygmaeus* results from the location of WPN on the shores of Szczecin Lagoon, which is subjected to an increasing input of nutrients from the polluted Odra River, leading to the development of hypereutrophic conditions [91]. Such conditions have presumably led to an exceptional abundance of aquatic nematoceran flies, especially chironomids [87], i.e., the predominant prey of *P. pygmaeus*.

We presume that the coincidence of spontaneous and active renaturation with the provision of surplus anthropogenic roosts and the abundance of prey due to contact with water subjected to eutrophication could create an optimal habitat for *P. pygmaeus*. These factors may also allow *P. pygmaeus* to outcompete, in tree roosts, other bat species that would, otherwise, benefit from the renaturation of the Park's forests but occupy much narrower ecological niches, such as *B. barbastellus*, as the latter is a rare example of a food specialist among Central European bats [92]. We cannot exclude the possibility that the success of *P. pygmaeus* affects even much more common and more generalist tree-dwelling bats that use different hunting tactics (*N. noctula*), reducing their relative abundance in the quantitative structure of the assemblages. If the affected species are characteristic of a particular habitat (e.g., *M. daubentonii* for water courses and lakes), then this increases the similarity of assemblages across habitats and might lead to the general depauperation of the bat fauna. If the proposed scenario is an accurate model, then the predominance of *P. pygmaeus* in all the studied forest habitats in WPN should be treated as an example of a hyperabundant native species, i.e., a taxon that was not introduced by humans but has benefited from their activities, negatively affecting other, less-successful native species [93]. A convincing field test for competitive exclusion between sympatric bat species would be hard to perform, while the sole reduction of the relative abundance of the other species could just be a statistical phenomenon, associated with no direct, negative effects on their population densities. However, the low number of localities in the case of the other taxa considered widespread and common, combined with only the marginal occurrence of old forest specialists, suggests otherwise.

## 5. Conclusions

Our study revealed a unique quantitative structure of a forest bat assemblage in a Central European protected area, characterized by an exceptionally high relative abundance of a single generalist species, *P. pygmaeus*, known for resistance to anthropopressure, flexible foraging strategy, and broad spectrum of roosts utilized, from the natural to the anthropogenic. This abundance has led to general depauperation of the bat fauna, with a low proportion of old forest specialists (the absence of *B. barbastellus* in summer), despite the high age of the tree stands and the availability of potential roosts. The most notable feature revealed via our survey is the high level of similarity in the structure of bat assemblages among all the sampled habitats, contrary to all the other forests of lowland Poland covered by our review. We suppose that *P. pygmaeus* may, in some ecological contexts, act as a hyperabundant native species, at least partially driven by anthropogenic cross-boundary subsidy; to our knowledge, it would be the first bat taxon revealing such a function. Climate change, eutrophication, and the urbanization of areas neighboring woodland patches may lead to similar situations in the future, reducing the potential benefits of forest renaturation for diversity among woodland bats. We admit that the hypothesis about the direct negative impact of *P. pygmaeus* on the abundance of the remaining species must remain speculative until a more convincing test is conducted, as the whole phenomenon could be purely

statistical. However, a uniformization this strong in the diversity and structure of bat assemblages among contrasting habitats remains a phenomenon worth paying attention to, unmatched even on a broader geographical scale.

**Supplementary Materials:** The following supporting information can be downloaded at: https://www.mdpi.com/article/10.3390/f15020337/s1, Table S1: Rules of reclassification of bat calls in cases when automatic classifiers provided identification pointing to the species either not occurring in the geographical region or being impossible to distinguish reliably due to a strong overlap in call parameters.; Table S2: Species composition of bats mist-netted in four different habitats in 29 forests in lowland Poland obtained from 33 publications and used for cluster analysis (Figure 7); Table S3: Species composition of bats mist-netted and recorded in different habitats in Wolin National Park during surveys in 2022–2023—the raw and adjusted data used in Figures 2–6 [94–124].

**Author Contributions:** Conceptualization: M.C.; methodology, M.C.; bat mist netting, handling, and identification, M.C., Z.W., K.B. (Katarzyna Borzym), E.J., J.B., M.J.-J. and K.B. (Konrad Bidziński); ultrasound recording, M.C., Z.W., K.B. (Katarzyna Borzym) and E.J.; automatic identification of bat call recordings, K.B. (Katarzyna Borzym), E.J., J.B. and K.B. (Konrad Bidziński); manual verification of recordings, M.C., Z.W. and K.B. (Katarzyna Borzym); preparation of databases, M.C., Z.W. and K.B. (Katarzyna Borzym); statistical analysis, M.C.; writing—original draft preparation, M.C.; writing—review and editing, M.C., Z.W., K.B. (Katarzyna Borzym), E.J., J.B., M.J.-J. and K.B. (Konrad Bidziński); data visualization, M.C.; map and spatial data, M.J.-J. and K.B. (Konrad Bidziński); preparation of references, M.J.-J.; project administration, M.C. and K.B. (Katarzyna Borzym) All authors have read and agreed to the published version of the manuscript.

**Funding:** This research was funded by the Forest Fund of State Forests National Forest Holding (*Fundusz Leśny Państwowego Gospodarstwa Leśnego Lasy Państwowe*), Grant No. EZ.0290.1.22.2022, through Contract No. 97/2022 between Wolin National Park (the recipient of the grant) and the Polish Society for Nature Protection 'Salamandra'.

**Data Availability Statement:** All data are provided in the Supplementary Material.

**Acknowledgments:** We would like to cordially thank the staff of Wolin National Park for their permission to conduct the study, organization of funding and accommodation, and help in selecting sites for mist netting and recording bat calls with Batcorders: Wioletta Nawrocka, Alicja Łepek, Konrad Wrzecionkowski, Radosław Skórkowski, Rafał Mackiewicz, Marek Szwarc, Wanda Janek, Andrzej Jabłonka, Jan Magda, Tomasz Bajor, Łukasz Potocki, Marcin Kądziołka, Artur Szymański, and Tomasz Kapral. We also thank all the participants of student research training camps in WPN in 2022–2023, who helped with bat mist netting: Emilia Czabrowska, Ewelina Janikowska, Amelia Rydzyńska, Magda Sitko, Arkadiusz Trzciński, Maja Ura, Maksymilian Wojtkiewicz, Barbara Komosińska, Wiktoria Chudzik, Paulina Kozłowska, Kacper Skokowski, Teresa Kowalewska, Rain Kosatko, and Blanka Kwiatkowska.

**Conflicts of Interest:** The authors declare no conflicts of interest.

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
