# Peer review of "Exceptionally Uniform Bat Assemblages across Different Forest Habitats Are Dominated by Single Hyperabundant Generalist Species"

_forests, doi:10.3390/f15020337_

Round 1

Reviewer 1 Report

Comments and Suggestions for Authors

Overall, this represents a very interesting study with an extensive and detailed set of field data that provide valuable new information on bat assemblages in Wolin National Park in Poland.

 However, there are a number of aspects of the interpretation of the data that are not appropriate.

First, and most important, the conclusion that the bat assemblages are uniform across different forest habitats is not supported by the data for two main reasons. First, the fact that one species is found commonly in all habitats does not mean that it has any effect on the other species. In fact, several of the other species do differ in abundance among habitats. Relative abundance is not an appropriate method for analysing impacts of species, because addition of large numbers of a new species will always reduce relative abundance of other species (for statistical reasons) even if they have no impact on the actual abundance of the other species. If P. pygmaeus did not occur at this site, you might find exactly the same numbers of all the other species. The conclusion that it is reducing the contribution of more specialized taxa is not appropriate.

Second, the four ‘habitats’ analysed in this study are all open habitats. The authors did not sample within the forest (e.g., below the canopy), nor did they differentiate vertical distribution of the bats (some species may be feeding high above the ground, while others may be very low). As mentioned in a few spots in the paper (but not the conclusions or the abstract), there is likely substantial vertical differentiation in habitat use among species, with species such as noctules making greater use of the open spaces high above the ground. Thus, the suggestion in the title and abstract of similarities among habitats is misleading, because all the habitats sampled were somewhat similar (i.e., open spaces) and did not differentiate where the bats were actually occurring within each sampled site. Also, the authors do not consider that, in fact, there were large differences in abundance among sites for each of the species considered.

The best way to address this concerns would be to present the data by analysing results WITHIN species instead of across species. This would reveal more accurately the strong differences among sites for some species. Thus, for example, Figures 2 and 4 should not be scaled by percentages. They should be scaled based on the absolute numbers of bats detected, or perhaps better yet, presented corrected for effort (number of bat detector nights or number of mist-net nights). From figure 4, for example, it is quite clear that the vast majority of Nyctalus noctula were detected on the shores of lakes and lagoons with only a handful detected in glades and canopy gaps. You would need to adjust for effort, but even so, it is clear there is a huge difference in their abundance in different habitats – quite different from the conclusion that the assemblages are uniform. This completely contradicts the conclusions in the abstract.

 The authors could also potentially enhance the value of this paper if they included some information on the feeding activity of bats in different areas, through analysis of feeding buzzes, assuming that feeding buzzes were detected regularly on their recorders. In most cases, it should be possible to identify feeding buzzes to species based on the approach calls that were detected before them. It would be a valuable addition to the paper to present information on how the different species were using each of these habitats for feeding.

Finally, I have a few concerns about some of the statistics. First, chi-square values need to be presented with their degrees of freedom to be interpretable. For example, I don’t know what was compared on line 234, but it seems rather odd that a chi-square value of 3336 would not be highly significant as it is much larger than all the other chi-square values. Second, chi-square is only valid for contingency analysis if the individual samples (i.e., each recording of a bat) are independent. However, this is not the case with these data for a few reasons. In particular, for the acoustic data, the same bat may be detected on multiple occasions; thus, the same bat may be counted multiple times in the sample which invalidates chi-square analysis. While this does not mean one can’t compare abundance among sites it does invalidate chi-square analysis. Also, adjusting for detectability further invalidates this assumption as some bats get counted multiple times (if they have low detectability each detection can turn into multiple bats).

Finally, while adjusting for detectability is conceptually an interesting approach, but it is only meaningful if the bats are using the same habitats within each sampled site. In fact, some species may be using different habitats within each site. In particular, the noctules may be foraging at much higher altitudes than the pipistrelles; adjusting their abundance for detectability loses this information. In any case, this adjustment is not necessary if the results are analysed within species, as is the most appropriate approach for looking at habitat segregation and usage.

Comments on the Quality of English Language

While the English is generally easy to understand, there are minor grammatical errors throughout the paper that could be corrected with appropriate copy-editing.

Author Response

Reviewer 1

Overall, this represents a very interesting study with an extensive and detailed set of field data that provide valuable new information on bat assemblages in Wolin National Park in Poland.

 However, there are a number of aspects of the interpretation of the data that are not appropriate.

>>First, and most important, the conclusion that the bat assemblages are uniform across different forest habitats is not supported by the data for two main reasons. First, the fact that one species is found commonly in all habitats does not mean that it has any effect on the other species. In fact, several of the other species do differ in abundance among habitats. Relative abundance is not an appropriate method for analysing impacts of species, because addition of large numbers of a new species will always reduce relative abundance of other species (for statistical reasons) even if they have no impact on the actual abundance of the other species. If P. pygmaeus did not occur at this site, you might find exactly the same numbers of all the other species. The conclusion that it is reducing the contribution of more specialized taxa is not appropriate.

We changed the final Conclusions and the last paragraph of Discussion to fit more Your objections, however we should still treat the possibility that the effect of predominant species on the other taxa is a causative relationship, not only a statistical effect, as at least plausible hypothesis. We agree that the absolute abundance of the other species do not necessarily have to be affected by a dominant species, even if their relative abundance is significantly reduced, however there are other issues which may support the hypothesis about the competitive exclusion. First, complete lack of old forest specialist, B. barbastellus, despite abundant, potentially preferred, habitat. The species is common in most of Poland and easily detectable, if nets are placed on forest roads. It is not the matter of reduced relative abundance (percentage) but of complete absence of particular taxon, which may be expected there and is usually met in such habitat (old deciduous forests rich in trees with exfoliating bark and crevices in trunks) in the whole biogeographical region. The idea of competitive exclusion of stenotopic, narrowly specialized, usually threatened, bat species by eurytopic, generalist and not-threatened member of that order is not a new one, we refer to it, when citing the paper of Arlettaz et al. (2000). ‘Reducing the contribution’ in the abstract, semantically refer to the reduced share in the assemblage (relative abundance, %). Finally, reduction of relative abundance (percentage) could be, indeed, purely numerical effect in case of extremely abundant, successful dominant, with no negative effect for the remaining taxa. Yet, very low frequency of occurrence (number of localities) of, otherwise common and widespread bat species (N. noctula, M. daubentonii) may suggest otherwise, however, increasing probability that even those species may be directly affected by commonness of P. pygmaeus. We admit that it would remain speculative if no convincing test for competitive exclusion is provided.

>>Second, the four ‘habitats’ analysed in this study are all open habitats. The authors did not sample within the forest (e.g., below the canopy), nor did they differentiate vertical distribution of the bats (some species may be feeding high above the ground, while others may be very low). As mentioned in a few spots in the paper (but not the conclusions or the abstract), there is likely substantial vertical differentiation in habitat use among species, with species such as noctules making greater use of the open spaces high above the ground. Thus, the suggestion in the title and abstract of similarities among habitats is misleading, because all the habitats sampled were somewhat similar (i.e., open spaces) and did not differentiate where the bats were actually occurring within each sampled site. Also, the authors do not consider that, in fact, there were large differences in abundance among sites for each of the species considered.

The studied habitats, even if their sets differ among the methods, are definitely NOT ALL OPEN. Forest roads in the study area are mostly (if not exclusively) covered by dense, multilayer tree canopy, thus, definitely, cannot be regarded as open-space. Moreover, nets located just within tree stands, near the roads, were lumped with those set across the roads, as they provided too small sample to treat them separately. So the discussed habitat class is called ‘forest roads and stand interiors’. Also the only streams sampled are flowing UNDER the canopy, while among the studied small ponds, one was located at small forest glade, while the two other were under the canopy again. We added that information into Materials and Methods, where we deal with distinguished habitat classes. Regarding our acoustic surveys, indeed, all recording points were located in at least semi-open situations (both aquatic and terrestrial) but, on contrary, all Batcorder points were located within the tree stands and most of them represented the closed habitats. As it appeared to be not clear enough (our fault!), we explained that in ‘Materials and methods’. Yes, it is true that species using different height about the ground may, in fact, use different microhabitats within the site (noctules flying even above the canopy) but it applies only for acoustic data, not mist netting, as bats were using exactly that height at which they were captured. In Discussion we added a comment about potential bias, resulting from the fact that some species may be recorded in one habitat, even if they use another one (with noctules as an example).

>>The best way to address this concerns would be to present the data by analysing results WITHIN species instead of across species. This would reveal more accurately the strong differences among sites for some species. Thus, for example, Figures 2 and 4 should not be scaled by percentages. They should be scaled based on the absolute numbers of bats detected, or perhaps better yet, presented corrected for effort (number of bat detector nights or number of mist-net nights).

In that particular paper, we focused not on habitat selection but on relative abundance and structure on bat assemblage. We suggest that usage of different habitats by particular species and structure of assemblage in particular habitats are two different phenomena, characterised by different set of parameters, even if they are interrelated (structure of assemblage is partially shaped by habitat preferences of species forming that assemblage). In the present paper we focused on the latter, but we acknowledge the potential in the analysis of the former; it would be, however, material for separate paper (we would probably prepare such a manuscript in future, based solely on acoustic data, as they provide the largest sample). Two main parameters are used to characterize structure of multi-species assemblage – one is diversity (we applied rarefaction curves for that purpose but it could be one of diversity indices), another is relative abundance (=dominance), i.e. % of individuals or biomass in the population, as applied by, e.g. Jones and Magurran (2018). We could perform additional analyses of the absolute abundance (therefore, habitat usage)  and expand both scope and size of the paper by approximately 1/3 but it would diminish its present consistency. Correction for effort is an idea worth applying but, we are afraid, it would be hard to obtain proper standardisation, as sometimes mist netting was interrupted by unexpected rain, while batteries in recording devices sometimes expired before sunrise due to extremely high level of bat activity. Relative abundance, as it does not require correction for effort, is not sensitive to such events.

>>From figure 4, for example, it is quite clear that the vast majority of Nyctalus noctula were detected on the shores of lakes and lagoons with only a handful detected in glades and canopy gaps. You would need to adjust for effort, but even so, it is clear there is a huge difference in their abundance in different habitats – quite different from the conclusion that the assemblages are uniform. This completely contradicts the conclusions in the abstract.

Still, P. pygmaeus is either dominant or co-dominant taxon in almost all samples across various habitats and methods, which, as revealed by our review analysis, is an unexpected result for, at least, the whole lowland Poland. Yes, there are secondary differences in relative abundance of particular species, associated with characteristics of that habitat (open water and glades – numerous N. noctula, open water and streams – M. daubentonii), but they do not change the general picture – still in every habitat, open or cluttered, dry or riparian/lacustrine, there is one species that occupies the first place in ‘the ranking of abundance’, which appeared to be a very rare (if not unique) case, at least among the areas sampled in Central Poland. And this is, what we state as final conclusion.

>>The authors could also potentially enhance the value of this paper if they included some information on the feeding activity of bats in different areas, through analysis of feeding buzzes, assuming that feeding buzzes were detected regularly on their recorders. In most cases, it should be possible to identify feeding buzzes to species based on the approach calls that were detected before them. It would be a valuable addition to the paper to present information on how the different species were using each of these habitats for feeding.

Yes, it would add another dimension to our study, however it relates more to the subject of habitat usage and habitat selection which we do not cover at al. What we focused on in that paper, is a structure of bat assemblages, i.e. relative abundance which may or may not result from individual preferences of the taxa forming the assemblage.

>>Finally, I have a few concerns about some of the statistics. First, chi-square values need to be presented with their degrees of freedom to be interpretable. For example, I don’t know what was compared on line 234, but it seems rather odd that a chi-square value of 3336 would not be highly significant as it is much larger than all the other chi-square values. Second, chi-square is only valid for contingency analysis if the individual samples (i.e., each recording of a bat) are independent. However, this is not the case with these data for a few reasons. In particular, for the acoustic data, the same bat may be detected on multiple occasions; thus, the same bat may be counted multiple times in the sample which invalidates chi-square analysis. While this does not mean one can’t compare abundance among sites it does invalidate chi-square analysis. Also, adjusting for detectability further invalidates this assumption as some bats get counted multiple times (if they have low detectability each detection can turn into multiple bats).

We agree with that suggestion, thus we decided to skip chi-square tests altogether. It could be more applicable to the mist netting data, but in that case the samples do meet another assumption of the test (minimal numbers in each cell of the table). We could partially lumped some ecologically similar and/or taxonomically related taxa but it does not always help (still zeroes or numbers <5 remain) and it not always make sense. We left only rarefaction and cluster analyses, as our data met the assumptions of that methods, as well as they point to our study goal more directly.

>>Finally, while adjusting for detectability is conceptually an interesting approach, but it is only meaningful if the bats are using the same habitats within each sampled site. In fact, some species may be using different habitats within each site. In particular, the noctules may be foraging at much higher altitudes than the pipistrelles; adjusting their abundance for detectability loses this information.

This is why we provided both adjusted and unadjusted results of acoustic survey. However, we added that dimension in Discussion, as we found that point valuable.

>>In any case, this adjustment is not necessary if the results are analysed within species, as is the most appropriate approach for looking at habitat segregation and usage.

Fully agree, but as mentioned above, we do not focus on habitat usage or selection but on the structure of bat assemblages, i.e. dominance of species forming those assemblages.

Literature cited:

Arlettaz, R.; Godat, S.; Meyer, H. Competition for food by expanding pipistrelle bat populations (Pipistrellus pipistrellus) might contribute to the decline of lesser horseshoe bats (Rhinolophus hipposideros). Biological Conservation 2000, Volume 93, Issue 1, pp. 55-60. DOI: https://doi.org/10.1016/S0006-3207(99)00112-3

Jones Faith A. M. and Magurran Anne E. 2018. Dominance structure of assemblages is regulated over a period of rapid environmental change. Biol. Lett.142018018720180187. http://doi.org/10.1098/rsbl.2018.0187

Reviewer 2 Report

Comments and Suggestions for Authors

The hypothesis posited by the authors is that different habitats should exhibit different bat assemblages. The authors go about testing this hypothesis at many sites using bat recorders and mistnetting within a national park. The conclusion of the authors is that the assemblage within the national park is "extremely uniform".  However, differences in the uniformity of the bat assemblage can be discerned within figure 2, figure 4, and figure 5. The authors argue that figure 3 and figure 6 support their conclusion citing the overlap of confidence intervals. The authors admit this at line274. I think other methods of testing this are more appropriate than the rarefaction plots. Furthermore, I think that saying that the assemblage of the national park is uniform is a product of the methods used. Many of the bats in this study have foraging ranges greater than 12 km. The distance between the furthest sites is around 20 km. These bats can forage across the entire park. The different types of sites are in too close proximity to discern a difference using the methods that the authors used. I think a chi-squared test between sites would have shown a difference in bat presence and activity. 

The authors then use a highly confusing figure (figure 9) to show that the sites of the national park cluster together. Please clearly state in the text how this figure should be interpreted. This figure is large and contains many pieces of information. I do not think this figure clarifies the matter of assemblage clustering or differences well enough to justify the page space that it occupies. 

All that said, I am lead to believe that the entire premise of the article relies on the hard to interpret figure 9. The take home that I get is that this national park is a nice place to live for the often less abundant Pipistrellus sp. 

I am unsure why two methods of autoID were used. Some calls were classified using Kaleidoscope while some were identified using bat ident. Please describe why multiple methods are used and on which data.  

Comments on the Quality of English Language

The introduction and the discussion appear to be well polished. However the methods and results use improper grammar. Many sentences are intelligible with effort but are difficult to read. Some sentences leave the point very ambiguous. 

Author Response

>>While the English is generally easy to understand, there are minor grammatical errors throughout the paper that could be corrected with appropriate copy-editing.

We went it through professional copy-editing to improve English.

>>The hypothesis posited by the authors is that different habitats should exhibit different bat assemblages. The authors go about testing this hypothesis at many sites using bat recorders and mist netting within a national park. The conclusion of the authors is that the assemblage within the national park is "extremely uniform".  However, differences in the uniformity of the bat assemblage can be discerned within figure 2, figure 4, and figure 5. The authors argue that figure 3 and figure 6 support their conclusion citing the overlap of confidence intervals. The authors admit this at line274. I think other methods of testing this are more appropriate than the rarefaction plots. Furthermore, I think that saying that the assemblage of the national park is uniform is a product of the methods used. Many of the bats in this study have foraging ranges greater than 12 km. The distance between the furthest sites is around 20 km. These bats can forage across the entire park. The different types of sites are in too close proximity to discern a difference using the methods that the authors used. I think a chi-squared test between sites would have shown a difference in bat presence and activity. 

Rarefaction curves were only to test for differences in DIVERSITY among habitats, and, if we know, it is the best way to do that, much better than Shannon-Wiener or Pielou indices, to which many concerns were raised recently, resulting in calls for abandonment of that measure altogether (Strong 2016). On contrary to the latter, rarefaction curves are considered valid method of diversity estimation (Gotelli and Graves 1996) and their use increase significantly in recent years. Overlap of confidence intervals in rarefaction curves is not the only line of evidence, however. First, we found that in every habitat sampled with every method, one and the same species occupies the highest position in the ranking of relative abundance, which appeared highly unusual, compared to any other forest, protected or unprotected, in lowland Poland.

Proximity of sites and commuting distances hardly matter in that case, as species use habitats based on preferred conditions (in case of bats – density of trees and undergrowth, openness of canopy, presence of open water surface, linear landscape elements) and that preferences differ strongly among species, usually associated with hunting tactics. Foliage gleaners usually fly in very cluttered conditions, aerial hawkers – not so much, although Pipistrellus fly much closer to obstacles than typical open-space, fast-flying species like Nyctalus, while water-surface foragers need just clutter-free water for hunting but can commute through very cluttered habitats as well. This is how species assemblage form and even sites located hundreds if not dozens of meters from each other can provide completely different species composition of the mist netted samples, even if they are located within home range of the same bat individuals or colonies. In most other forests used in cluster analysis (fig. 9) most sites were also located no more that few kilometres from each other (sometimes less than 1 km), yet, if they represented drastically different habitat (e.g. river vs. small pond, lake vs. forest road), they yielded bat samples of completely different taxonomic structure and that characteristics was consistent within similar habitats across long geographical distances.

And, judging from radiotracking data, the dominant species does not commute on such large distances. During lactation period, P. pygmaeus visit foraging sites located no further than 2 km, with median MCP span about 2 km as well, as median MCP about 170 ha; slightly higher values are typical for its sibling, P. pipistrellus (Davidson-Watts and Jones 2004). Experimental exclusion of P. pygmaeus colonies confirmed that picture – removed individuals moved to alternative roosts located no further than 1.5 km and commute to foraging habitats located 1.5 km on average as well, with mean area of core range only 46 ha (Stone et al. 2016). So definitely, those bats cannot forage in the whole 10937 ha of the park, with 20 km distance between the furthest sites. The remaining species are known to commute only slightly further: P. nathusii (maximum foraging distances usually 3-4 km, exceptionally up to 6.6 km, Flaquer et al. 2009), M. nattereri (3 km on average, home range 5-500 ha, Smith & Racey 2008), M. daubentonii (up to 6,3 km, Dietz et al. 2006), P. auritus (less than 3 km, Swift 1998) and even fast-flying, open-space forager, N. leisleri (less than 6 km, with home range up to 1800 ha, Waters et al. 1999). Only two species, regularly netted and recorded in the Park really cover such large distances as You claim, these are E. serotinus (up to 12 km, Catto et al. 1996) and N. noctula (up to 23 km, but foraging sites were located no further than 6,3 km, Mackie and Racey 2007). Ironically, the latter two species, with the longest commuting distances and largest home ranges, thus expected to get everywhere during one night after emerging from the roost, were netted only at small minority of sites (4 and 5 of 17, respectively), while the species with quite limited movement capacities appeared to be the most widespread (tab. 1). It further falsifies the hypothesis about a close distance between sites as a causative agent behind high similarity among compared habitat..

>>The authors then use a highly confusing figure (figure 9) to show that the sites of the national park cluster together. Please clearly state in the text how this figure should be interpreted. This figure is large and contains many pieces of information. I do not think this figure clarifies the matter of assemblage clustering or differences well enough to justify the page space that it occupies. All that said, I am lead to believe that the entire premise of the article relies on the hard to interpret figure 9. The take home that I get is that this national park is a nice place to live for the often less abundant Pipistrellus sp. 

We cannot agree with that, at least not completely. Figure 9 is an important piece of analysis, placing the WPN in the broader context, showing that, indeed, predominance of one and the same species in all habitats sampled with mist netting is a specific feature of WPN, which was not revealed in any other forest studied in Poland. So, it rather supplements the line of evidence, provided by earlier data and their visualisation in fig. 2-7 and tab. 1 – that, no matter what habitat do You sample in the Park and no matter what method do You apply, there is always one species that predominate in the assemblage (and that species is P. pygmaeus). However, until we do not provide a review of data from other forests (which is not available in any earlier paper – we had to review all that petty, local contributions in one analysis), we cannot prove if that situation is something unusual or rather common in that part of Palearctic. Using fig. 9, we prove the former – usually, in central Europe, when You mist net bats over forest river, You get fundamentally different species composition than at forest roads, while small standing water bodies, like forest pools, provide completely different species composition from forest rivers as well. That fig. 9 provides additional quality to our paper, as it makes – maybe crude but useful – review and numerical analysis of several dozen papers in local, often Polish-language, journals, hardly available for broader audience. It is the only recollection of such huge amount of data, gathered during local bat surveys in Polish forests (contrary to Nearctic and Neotropics, bat mist netting data rarely occur in international, peer-reviewed literature), so it adds merit to our paper, not only it acts as a test of our hypotheses. Stil, cluster trees are extremely common tool for data visualisation in various fields of biology for a long time, while heat maps are of growing popularity in recent years, so we do not afraid that fig. 9 will be highly confusing or hard to interpret for most readers.

Additionally, the picture presented in that analysis appears to be resistant to choice of statistics. Grouping of all WPN habitats together remained stable, whatever linkage method or measure of distance we chose. This is why we provide raw data for that analysis as Supplementary Materials, so everyone can check it, applying various clustering methods. And, in all cases, that tight clustering of all WPN habitats is a unique feature of that park. Most often, contrasting habitats in the same forest are usually dispersed across the whole cladogram. So, we suppose it is a strong piece of evidence.

>>I am unsure why two methods of autoID were used. Some calls were classified using Kaleidoscope while some were identified using bat ident. Please describe why multiple methods are used and on which data.  

We described that already in Methods – our recordings from EM3+, Pettersson D-1000X and MiniBat were analysed in Kaleidoscope (software originally created for EM3+ or any other Wildlife Acoustic recorder), while the recordings we obtained from the Park staff and conducted with Batcorder recorders were analysed in batIdent (with manual verification in bcAnalyze). The reason for that discrepancy is the fact, that most available recording devices record bat calls in WAV format (compatible with Kaleidoscope software), while Batcorders record bat calls in RAW format that can be neither opened nor processed with Kaleidoscope. Therefore, we needed special software, dedicated solely to Batcorder recordings.

Comments on the Quality of English Language

>>The introduction and the discussion appear to be well polished. However the methods and results use improper grammar. Many sentences are intelligible with effort but are difficult to read. Some sentences leave the point very ambiguous. 

We improved language by sending the text to the professional proof-reading service.

Literature cited:

Catto, C. M. C., Hutson, A. M., Racey, P. A., and Stephenson, P. J. 1996. Foraging behaviour and habitat use of the serotine bat (Eptesicus serotinus) in southern England. J. Zool. 238: 623-633.

Davidson-Watts, I. and Jones, G. (2006), Differences in foraging behaviour between Pipistrellus pipistrellus (Schreber, 1774) and Pipistrellus pygmaeus (Leach, 1825). Journal of Zoology, 268: 55-62. https://doi.org/10.1111/j.1469-7998.2005.00016.x

Dietz, M., Encarnação, J. A., and Kalko, E. K. V. 2006. Small-scale distribution patterns of female and male Daubenton’s bats (Myotis daubentonii). Acta Chiropter. 8: 403-415.

Flaquer C, Puig-Montserrat X, Goiti U, Vidal F, Curcó A, Russo D (2009) Habitat selection in Nathusius’ pipistrelle (Pipistrellus nathusii): the importance of wetlands. Acta Chiropterol 11:149–155

Gotelli, Nicholas J. and Graves, Gary R. 1996. Null Models in Ecology. Washington, D.C.: Smithsonian Institution Press.

Mackie I., Racey P. 2007. Habitat use varies with reproductive state in noctule bats (Nyctalus noctula): implications for conservation. Biol. Conserv. 140: 70-77.

Smith P. G., Racey P. A. 2008. Natterer’s bats prefer foraging in broad-leaved woodlands and river corridors. J. Zool. 275: 314-322.

W.L Strong, Biased richness and evenness relationships within Shannon–Wiener index values, Ecological Indicators, Volume 67, 2016, Pages 703-713

Swift S. E. 1998. Long-eared Bats. T & AD Poyser Natural History, 182 pp.

Stone, Emma & Zeale, Matt & Newson, Stuart & Browne, William & Harris, Stephen & Jones, Gareth. (2015). Managing Conflict between Bats and Humans: The Response of Soprano Pipistrelles (Pipistrellus pygmaeus) to Exclusion from Roosts in Houses. PloS one. 10. e0131825. 10.1371/journal.pone.0131825.

Waters D., Jones G., Furlong M. 1999. Foraging ecology of Leisler’s bat (Nyctalus leisleri) at two sites in southern Britain. J. Zool. 249: 173-180.

Reviewer 3 Report

Comments and Suggestions for Authors

This paper presents a very interesting study, and results which are unexpected and of high importance. However, there are a few areas which need improvement: (1) Mist-netted bats were identified and released, and apparently no voucher specimens were collected. Because many congeneric bat species (including vespertilionids) are externally quite similar and difficult to distinguish and identify confidently, the authors need to convince the reader that they could indeed identify all species captured in this study--simply citing Dietz and Kiefer isn't convincing. (2) In lines 120-121 there is mention of some nights not sampled due to weather conditions, but there is no indication of how many nights were not sampled or more to the point, how many nights were included in the study. (3) The paragraph of lines 149-159 lists some species pairs which could not be confidently distinguished by sonar recordings and were therefore lumped in the analyses, but I got lost trying to remember which pairs these were, in reading the tables and figures which followed in the manuscript. A brief table summarizing which species were "lumped" might be useful for this. (4) In line 205 it says that 455 bats were captured, but it isn't clear whether this was actual capture (mist-net) or all "captures" including sonar. Please clarify.

Comments on the Quality of English Language

The English usage must be improved. A native-speaking English person with an academic background (preferably biology) must carefully review this manuscript before it is resubmitted. I realize that this may be difficult and time-consuming to arrange, but it is important. As it is, the incorrect English usage detracts from the paper, which otherwise will be a valuable contribution to the European bat community literature, given attention to the suggestions indicated above.

Author Response

Reviewer 3

This paper presents a very interesting study, and results which are unexpected and of high importance. However, there are a few areas which need improvement:

>>(1) Mist-netted bats were identified and released, and apparently no voucher specimens were collected. Because many congeneric bat species (including vespertilionids) are externally quite similar and difficult to distinguish and identify confidently, the authors need to convince the reader that they could indeed identify all species captured in this study--simply citing Dietz and Kiefer isn't convincing.

In fact, all species of bats that were mist netted by us or expected to be mist netted by us, as known to occur in North-Western Poland, are well known to be distinguishable in hand, using several morphological features, which combinations are completely non-overlapping among particular taxa – in fact those morphological identifications are much less ambiguous than acoustic ones. Using those combinations is a mandatory basics in training for any Polish bat worker, including our students. We cited Dietz and Kiefer’s handbook only because it is the most recent and comprehensive literary source for all that diagnostic features and their combinations – a source that we, indeed, kept at hand during our field work on Wolin in 2022-2023. In fact, it depends of the geographical origin of the reader if they are convinced or not. I assure You that those working in Central and Northern Europe or North America and having an experience with depauperate bat fauna of that regions, will be easily convinced (or even will not question those IDs, if any potential doubts may arouse, they would be around bioacoustics), because all of them identify a dozen or so local species (often less than 10), even the most similar one, on a daily bases. Those working in Southern Asia, equatorial East Africa or Neotropics, when bat faunas are extremely diverse (especially in Oriental region, with abundance of vespertilionids, including still undescribed species), will doubt that pipistrelles or Myotis can be all light-heartedly released after capture, as their ID often requires examination of teeth and skull. It could still pose some problems in Mediterranean zone of Europe. This is, however, not the case in Poland or any other temperate zone European country, at least until early 20., when knowledge of external identification features increased. Guidelines to conduct bat surveys in The United Kingdom (with bat fauna including all the species recorded on Wolin National Park), when describing procedure of mist netting, never recommend to collect voucher specimens, simply because it is required for persons that obtained license for capturing bats, to recognize all of them in hand. Moreover, they recommend to release all captured bats, among which there are plenty of Myotis and Pipistrellus, as fast as possible, ideally within minutes of being captured (Collins 2016). In fact, Dietz and Kiefer should be convincing as a reference, as any ID feature recommended by that book for identification of the species occurring in northern Poland requires no sacrificing of captured bats, collecting specimens or skull preparation. We add that information in Material and Methods to eliminate any doubts in that field. Other, recent European studies that deal with similar set of species were also based on hand identification and release of living individuals captured in mist nets (e.g. Vlaschenko et al. 2022).

>>(2) In lines 120-121 there is mention of some nights not sampled due to weather conditions, but there is no indication of how many nights were not sampled or more to the point, how many nights were included in the study. (3) The paragraph of lines 149-159 lists some species pairs which could not be confidently distinguished by sonar recordings and were therefore lumped in the analyses, but I got lost trying to remember which pairs these were, in reading the tables and figures which followed in the manuscript. A brief table summarizing which species were "lumped" might be useful for this. (4) In line 205 it says that 455 bats were captured, but it isn't clear whether this was actual capture (mist-net) or all "captures" including sonar. Please clarify.

(2) The number of nights (18) has been already given in line 119, these were exclusively sampled nights and its equal to the number of nights included in the study. During our stay in Wolin National Park there was only one night with extremely high wind speed, when we did not attempt to set mist nets, not only because it would not provide any comparable data but also for safety reasons. One night was interrupted by unsuspected light rain just about noon, therefore, in accordance with our protocol (no netting during rain), we stopped netting at that time, yet it appeared quite fruitful (41 individuals of 4 species, 6 nets, i.e. 6.8 individuals/net). As we focused on the structure of assemblage, not activity, so effort really did not matter, we included that night in the study. The lines 120-121 did not say that some nights were not sampled due to weather conditions but that the nights when we did sample, were characterised by particular weather conditions. This is not exactly the same. (3) OK, but we put that table in the Supplementary Materials to retain recent structure of the main text, as it is rather secondary technical matter, (4) ‘captures’ means ‘captures’, by definition bats that fell in mist nets, not ‘recordings’, ‘calls’, nor ‘echolocation call sequences’, which would refer to ultrasound detection. We clarified that in the text.

Comments on the Quality of English Language

>>The English usage must be improved. A native-speaking English person with an academic background (preferably biology) must carefully review this manuscript before it is resubmitted. I realize that this may be difficult and time-consuming to arrange, but it is important. As it is, the incorrect English usage detracts from the paper, which otherwise will be a valuable contribution to the European bat community literature, given attention to the suggestions indicated above.

We used professional proof-reading service to improve our English, thank You for that comment.

Literature cited:

Collins J. (ed.) 2016. Bat Surveys for Professional Ecologists: Good Practice Guidelines (3rd edn). The Bat Conservation Trust, London.

Vlaschenko, A.; Kravchenko, K.; Yatsiuk, Y.; Hukov, V.; Kramer-Schadt, S.; Radchuk, V. Bat Assemblages Are Shaped by Land Cover Types and Forest Age: A Case Study from Eastern Ukraine. Forests 2022, Volume 13, no. 10: 1732. DOI: https://doi.org/10.3390/f13101732